# Effect of Monosodium Glutamate on Saltiness and Palatability Ratings of Low-Salt Solutions in Japanese Adults According to Their Early Salt Exposure or Salty Taste Preference

**DOI:** 10.3390/nu13020577

**Published:** 2021-02-09

**Authors:** Rieko Morita, Masanori Ohta, Yoko Umeki, Akiko Nanri, Takuya Tsuchihashi, Hitomi Hayabuchi

**Affiliations:** 1Graduate School of Health and Environmental Sciences, Fukuoka Women’s University, Fukuoka 813-8529, Japan; r.morita@fwu.ac.jp (R.M.); umeki@fwu.ac.jp (Y.U.); nanri@fwu.ac.jp (A.N.); 2Steel Memorial Yawata Hospital, Kitakyushu, Fukuoka 805-8508, Japan; takuya.tuti@gmail.com; 3Department of Food Science and Nutrition, Faculty of Human Life and Environment, Nara Women’s University, Nara 630-8506, Japan; h.hayabuchi@cc.nara-wu.ac.jp

**Keywords:** salt reduction, umami, palatability, taste preference, generation, regional difference

## Abstract

Using umami can help reduce excessive salt intake, which contributes to cardiovascular disease. Differences in salt-exposed environment at birth and preference for the salty taste might affect the sense of taste. Focusing on these two differences, we investigated the effect of monosodium L-glutamate (MSG) on the saltiness and palatability of low-salt solutions. Japanese participants (64 men, 497 women, aged 19–86 years) tasted 0.3%, 0.6%, and 0.9% NaCl solutions with or without 0.3% MSG to evaluate saltiness and palatability. They were also asked about their birthplace, personal salty preference, and family salty preference. Adding MSG enhanced saltiness, especially in the 0.3% NaCl solution, while the effect was attenuated in the 0.6% and 0.9% NaCl solutions. Palatability was rated higher with MSG than without MSG for each NaCl solution, with a peak value for the 0.3% NaCl solution with MSG. There was no difference in the effect of umami ingredients on palatability between the average salt intake by the regional block at birth and salty preference (all *p* > 0.05). Thus, adding an appropriate amount of umami ingredients can facilitate salt reduction in diet while maintaining palatability regardless of the salt-exposed environment in early childhood or salty preference.

## 1. Introduction

The importance of sodium reduction as a practical prevention measure for cardiovascular disease is widely known [1,2]. However, salt intake in almost all countries exceeds the World Health Organization’s (WHO’s) recommendation of <5 g/day [3]. High salt intake has been reported as a leading dietary risk factor in over 3 million deaths and 70 million disability-adjusted life years in 2017 [4]. Salt reduction is an urgent issue worldwide [5].

In Japan, the National Health and Nutrition Survey in 2019 reported that the salt intake of Japanese adult men was 10.9 g/day and that of adult women was 9.3 g/day [6]. These amounts are approximately twice the amount recommended by the WHO. Since the 1950s, stroke has been the leading cause of death in Japan [7]. Tomonari et al., in their investigation of the mutual relationship among salt intake, blood pressure, and stroke mortality in 12 regions of Japan, reported that salt intake was an independent factor for stroke mortality [8]. Salt intake has been found to be higher in Tohoku than in other regions of Japan; indeed, in 1980, it accounted for the highest salt intake (15.8 g/day, Table 1). The intake of miso and pickles has also been reported as higher in Tohoku than in other regions [9].

Takachi et al. reported that self-reported taste preferences for miso soup were significantly associated with 24-h urinary sodium excretion and daily salt intake [10]. Food preferences established in early childhood continue into later life [11]. Salt preference is affected by the dietary habits of pregnant mothers and experiences with food during the first year of life [11,12]. Therefore, those who experienced high salt exposure during early childhood may have a high salty taste preference. Moreover, salty taste preference can also be influenced by the family’s salt use habits [13,14]. Many studies have reported that the use of glutamate is an effective way to reduce salt intake while maintaining the palatability of food [15,16]. Soup is a common dish worldwide, and miso soup is a daily food in Japan. It has been reported that glutamates such as monosodium glutamate (MSG) and calcium diglutamate (CDG) enhance the palatability of low-salt soups when used as a solvent, such as in clear soup, pumpkin soup, and chicken broth, and contribute to salt reduction [17,18,19]. Umami taste perception varies significantly among individuals. The differences in sensitivity can result from genetic variations in taste receptors [20], familiarity with umami [21,22], or hormonal levels [23]. Furthermore, umami taste perception can be enhanced by repeated exposure [20,24]. The interaction between umami and salt perception, however, remains unclear.

The effect of using MSG on low-salt solutions can be influenced by the salt exposure environment involving early childhood and family salty preference. However, no studies have investigated the relationship of the effect of umami in low-salt solutions with salt exposure environment in early childhood and salty taste preference. In this study, we focused on the differences between salt exposure environment in early childhood and current salty taste preference, and aimed to investigate the effect of adding sodium glutamate on the saltiness and palatability of low-salt solutions.

## 2. Materials and Methods

### 2.1. Study Design

This study was conducted from July 2017 to November 2018 in Japan. Sensory evaluations were performed and a questionnaire survey was administered at eight universities and 11 health seminars in 13 prefectures of Japan (Aomori, Miyagi, Tokyo, Chiba, Saitama, Kanagawa, Shizuoka, Nara, Hiroshima, Fukuoka, Nagasaki, Kagoshima, and Okinawa). The study protocol and material have been described in detail elsewhere [25].

These experiments were approved by the Research Ethics Committee at Fukuoka Women’s University (No. 2016-31) for students and the Research Ethics Review Committee at Nara Women’s University (No. 18-02) for attendees of health seminars. All participants provided signed informed consent prior to the study. The approved experiments were registered in the University Hospital Medical Information Network Clinical Trials Registry (UMIN000035280 and UMIN000035289).

### 2.2. Participants

The participants were 259 students from eight universities, and 392 attendees from 11 health seminars. From a total of 651 participants, we excluded 20 participants who had a taste disorder, eight participants who did not answer the question on taste disorder in the questionnaire, and 40 participants who did not complete the sensory evaluation test. In addition, we excluded 14 participants who did not answer the questionnaire, and eight participants who were not from Japan. Ultimately, 561 participants (64 men and 497 women) were included in the analysis. The flow chart of the participants is shown in Figure 1.

### 2.3. Sensory Evaluation

Previously, we examined the saltiness, umami, and palatability of 48 aqueous solutions containing eight different concentrations of NaCl (0.2, 0.3, 0.4, 0.5, 0.6, 0.7, 0.8, and 0.9%) and six different concentrations of MSG (0.1, 0.2, 0.3, 0.4, 0.5, and 0.6%) by sensory evaluation among female students and teachers [26]. Based on the results of the previous examination, six samples were prepared, which included 0.3%, 0.6%, and 0.9% NaCl solutions with or without 0.3% MSG. Saltiness and palatability ratings were assessed using a visual analogue scale (VAS) [27]. The VAS scale represented a minimum rating at the left end (not at all salty or extremely unpalatable) and a maximum rating at the right end (extremely salty or palatable) for each sample. Each solution was tested twice, with the order of the samples being changed after 15 min or more. The participants were asked to rinse their mouths with water before and after each sample evaluation.

### 2.4. Questionnaire Survey

Using a questionnaire, the participants were inquired about their individual characteristics of sex, age (years), smoking habit (current, former, and never), use of medication (yes and no), and birthplace. In addition, they were inquired about the degree of personal salty taste preference (very light, light, middle, strong, and very strong), degree of family salty taste preference (very light, light, middle, strong, and very strong), and degree of salt reduction efforts (always, sometimes, rarely, and never).

### 2.5. Average Salt Intake by Regional Block at Birth

In the present study, the average salt intake by regional block at birth was defined as the environmental indicator of salt exposure in early childhood. The average salt intake was obtained from the results of the National Nutrition Survey from 1980 to 2000, which has been conducted every year since 1946. There were no data on the average salt intake by regional block prior to 1979. We adopted the data of salt intake in 1980, as the salt intake might have been higher before 1980 than after 1980, according to annual changes in the salt intake of the Japanese from 1972 to 1980 [9]. Participants were divided into three groups based on generations: the born-in-the-2000s group (from 19 to 20 years old), the born-in-the-1990s group (from 21 to 30 years old), and the born-before-the-1980s group (over 31 years old). After that, the average salt intake data for 2000, 1990, and 1980 was used for the born-in-the-2000s group, the born-in-1990s group, and the born-before-1980s group, respectively. The birthplaces were also divided into 12 regional blocks according to the National Nutrition Survey in Japan (Table 1). Eventually, the averages of salt intake by regional block at birth in 1980, 1990, and 2000 were obtained for the 12 regional blocks.

### 2.6. Statistical Analysis

The average salt intake by regional block at birth was coded using the tertile as low (less than 12.6 g/day), middle (over 12.6 g/day and less than 13.4 g/day), and high (over 13.4 g/day). Moreover, degrees of personal and family salty taste preference were categorized into three groups: light (very light and light), middle, and strong (very strong and strong). The first and second taste evaluations of the saltiness and palatability of the six solutions were averaged. Chi-square tests were used to analyze the difference between sex and age groups. The normality of data distribution was tested using the Shapiro–Wilk test, and the distribution was not normal. Normal distribution methods were used in this study, because the statistical power was set such that VAS ratings were not concentrated around either extreme of the scale [28]. Repeated measures analysis of variance (ANOVA) was used to compare saltiness or palatability ratings of the six solutions. Then, pairwise comparisons were made using Tukey’s honestly significance difference procedure. In addition, analysis of covariance (ANCOVA) was employed to compare the saltiness or palatability ratings of each group based on sex, age group, salt intake by regional block at birth, personal salty taste preference, and family salty taste preference for the six solutions adjusted for sex, age group (19–20 years, 21–40 years, and ≥41 years), smoking habit (current, former, and never), salt reduction efforts (always, sometimes, and rarely/never), and use of medicine (yes and no). Statistical significance was defined as *p* < 0.05. Statistical analysis was conducted using JMP statistical software (JMP Pro 15.1.0, SAS Institute Inc., Cary, NC, USA).

## 3. Results

### 3.1. Characteristics of the Participants

Table 2 presents the characteristics of the participants by sex and age groups. The majority of the participants were female (88.6%) and never smokers (92.5%). The percentage of participants who were over 41 years old was 54.2% (mean ±SD: 45.45 ± 23.99). The percentage of participants with a high salt intake by regional block at birth was higher for the over 41-year-old age group than for the other age groups. Regarding salt reduction efforts, 72.5% participants answered that they engaged in such efforts “always” or “sometimes” (approximately 50% of the over 41-year-old age group). Thirty-four percent of the participants were taking medications, and this was particularly noticeable in the over 41-year-old age group (87.4%).

### 3.2. Saltiness

The means and standard errors (SEs) of saltiness VAS ratings by sex, age groups, the three levels of salt intake by regional block at birth, and the three levels of salty taste preference are shown in Table 3. Repeated measures ANOVA results showed significant differences in the saltiness ratings of the six solutions among all groups based on sex, age groups, the three levels of salt intake by regional block at birth, and the three levels of salty taste preference (all *p* < 0.001). The higher the NaCl concentration, the higher was the saltiness rating. The post-hoc test results showed that the 0.3% NaCl solutions with MSG showed significantly higher saltiness ratings than those without MSG in all groups (all *p* < 0.05). The 0.6% NaCl solution with MSG showed a significantly higher rating than the solution without MSG in the 21–40-years-old age group (*p* < 0.05). ANCOVA results showed that there were significantly different ratings for each NaCl solution with and without MSG among the three age groups, except for the rating for the 0.3% NaCl solution alone (0.3% NaCl alone, *p* = 0.304; other solutions, *p* < 0.001), with its lowest ratings being in the over 41-years-old age group. As for salt intake by regional block at birth, there was a significantly different rating for the 0.9% NaCl solution with MSG among the three groups (*p* = 0.008), and the saltiness ratings of the low-level group were higher than those of the other groups for all solutions. Additionally, with regard to personal and family salty taste preference, there were significantly different ratings for the 0.3% NaCl solution without MSG among the three groups (personal salty taste preference, *p* = 0.002; family salty taste preference, *p* = 0.028), and the ratings of the strong groups for the 0.3% NaCl solution without MSG were lower than those of other groups.

### 3.3. Palatability

The means and SEs of palatability VAS ratings by sex, age groups, the three levels of salt intake by regional block at birth, and the three levels of salty taste preference are shown in Table 4. Repeated measures ANOVA results showed significant differences in the average ratings of the six solutions among all groups based on sex, age groups, the three levels of salt intake by regional block at birth, and the three levels of salty taste preference (all *p* < 0.001). The results of the post-hoc test showed that the 0.3%, 0.6%, and 0.9% NaCl solutions with MSG showed significantly higher ratings for palatability than those without MSG among all groups (all *p* < 0.05), except for the rating of the 0.9% NaCl solution in the male group (*p* = 0.069). Regardless of age groups, salt intake by regional block at birth, and salty taste preference, adding MSG significantly enhanced the solutions’ palatability. The palatability ratings of the 0.3% NaCl solution with MSG were almost twice as high as those of the 0.3% NaCl solution without MSG in all groups. Moreover, the palatability ratings of the 0.3% NaCl solution with MSG were the highest among the six solutions in the female group, while there was no significant difference between the rating of the 0.3% NaCl solution with MSG and 0.6% NaCl solution with MSG in the male group, the three age groups, the three levels of salt intake by regional block at birth, and the three levels of salty taste preference. ANCOVA results showed significantly different ratings for palatability among age groups for the 0.6% NaCl solution without MSG (*p* = 0.040), with the rating of the over 41-years-old age group being lower than that of the other two groups. There were significantly different ratings for palatability among the three levels of personal salty taste preference for the 0.9% NaCl solution with MSG (*p* = 0.007), with the rating of the light group being lower than that of the other groups. Additionally, peak ratings of palatability were expressed for the 0.3% NaCl solutions with MSG, and there were no significant differences in palatability ratings among the solutions having peak ratings in each group (sex, *p* = 0.273; age groups, *p* = 0.147; salt intake by regional block at birth, *p* = 0.642; personal salty taste preference, *p* = 0.624; family salty taste preference, and *p* = 0.989). Among the 0.3%, 0.6%, and 0.9% NaCl solutions without MSG, the 0.6% NaCl solution had the highest palatability rating in all groups except for the male group, which had a peak rating for the 0.9% NaCl solution. Among the 0.3%, 0.6%, and 0.9% NaCl solutions with MSG, the 0.3% NaCl solution had the highest rating across all groups. Moreover, all solutions with MSG had peak ratings at a lower NaCl concentration than the solutions without MSG, regardless of sex, age, levels of salt exposure environment in early childhood, and salty taste preference.

## 4. Discussion

This study demonstrated that MSG enhanced the palatability of low-salt solutions, regardless of sex, age, salt intake by regional block at birth, and salty taste preference. Additionally, the 0.3% NaCl solution with MSG showed peak values of palatability ratings regardless of sex, age, salt intake by regional block at birth, and salty taste preference. This is the first study to investigate the effect of MSG, that is, umami ingredients, on low-salt solutions while considering the difference between the salt exposure environment in early childhood and current salty taste preference.

In a previous study, we presented the results of 584 participants evaluating six solutions (0.3%, 0.6%, and 0.9% NaCl solutions with or without 0.3% MSG); the results suggested that MSG enhanced the palatability of low-salt solutions regardless of sex, age, region, smoking habit, two hours of fasting, and medication [25]. The present study investigated the effect of MSG on low-sodium solutions, based on the variables of sex, age, levels of salt exposure environment in early childhood, and on salty taste preferences to contribute to the generalization of the effects of MSG.

In this study, it was shown that saltiness ratings depended on NaCl concentration of the solution, while palatability ratings were independent of NaCl concentration and got the peak value at a lower NaCl concentration (0.3% NaCl with 0.3% MSG) than the general concentration (around 1.0% NaCl) at which Japanese individuals consumed soup. In the previous studies investigating the interaction of NaCl and umami (MSG or CDG) in different types of soups, it was shown that the saltiness depended on NaCl concentration of the solution, while palatability ratings appeared parabolic wherein the peaks were observed at a different, medium-salty NaCl concentration [17,18,19]. Our results were broadly consistent with these previous reports.

A significant enhancing effect was observed for saltiness by adding MSG in the 0.3% NaCl solution regardless of sex, age, salt intake by regional block at birth, and salty taste preference. Although the detailed mechanism has not been clarified, it is known that saltiness sensitivity decreases with aging [29,30]. Barragan et al. conducted an evaluation test on participants aged 18–80 years, and reported that the 37–50-year-old and 51–80-year-old groups had a significantly reduced salty taste compared to the 18–36-year-old group [31]. In this study, the saltiness ratings of the over 41-years-old age group were lower than those of the other two younger age groups, a trend consistent with previous studies [29,30,31]. For the level of salt intake by regional block at birth, the saltiness ratings of the low-level group for the 0.9% NaCl solution with MSG were higher than those of the other groups. This means that those who were exposed to low levels of salt in early childhood were more sensitive to salty taste than those exposed to high salt levels. These results support previous studies reporting that taste preferences established under the influence of food experiences in early childhood continue through life [11,12].

In our previous study, MSG enhanced ratings of the 0.3%, 0.6%, and 0.9% NaCl solutions on palatability, and the 0.3% NaCl solution with MSG showed the highest palatability ratings. In addition, we suggested that it might be possible for the 0.3% NaCl solution with MSG (Na: 0.391 g/mL) to have a reduction of approximately 60% sodium compared to the 0.9% NaCl solution with MSG (Na: 0.155 g/mL) without a loss of palatability [25]. In the present study, MSG significantly enhanced the palatability ratings of the 0.3%, 0.6%, and 0.9% NaCl solutions among all groups, with the exception of the 0.9% NaCl solution in male. Moreover, the rating for the 0.3% NaCl solution with MSG showed the highest palatability regardless of sex, age, salt intake by regional block at birth, and salty taste preference. The enhanced effect of MSG was the same in the over 41-year-old age group with a weakened salty perception as in the other two younger age groups. As for palatability in males, the 0.9% NaCl solution obtained the peak rating among the three solutions of NaCl without MSG, and this rating was higher than that given by female and groups based on other classifications. While the sense of palatability in males might be low, the peak rating for palatability shifted from the 0.9% to 0.3% NaCl solution due to the addition of MSG. This indicates that adding MSG might be effective in reducing salt intake in males.

In this study, we investigated the influence of salt exposure environment in early childhood on the palatability using MSG on low-salt solutions. Our results showed that early childhood salt exposure did not affect enhancement of palatability using MSG in low-salt solutions. Kobayashi et al. have shown that sensitivity to umami taste is largely dependent on familiarity with umami taste [21]. As the Japanese participants had extensive experience with MSG in Japanese food, they might have been sensitive to the effect of umami ingredients on palatability.

We also assessed the influence of the degree of salty taste preference of individuals and families on sensory evaluation results. Regarding palatability, the ratings of the 0.3% NaCl solution with MSG showed a peak in all groups regardless of the degree of salty taste preference. Comparing the peak ratings among the strong, middle, and light groups, no significant difference was found. Uechi et al. examined urinary sodium excretion among participants in 47 prefectures of Japan and reported no domestic fluctuation [32]. In 1980, there was a difference in salt intake of about 5 g/day by regional block in Japan, from the highest (Tohoku: 15.8 g/day) salt-consuming region to the lowest (Kinki-1: 10.9 g/day) (Table 1). However, in 2018, the difference was only 1.6 g/day (the highest salt-consuming region, Tohoku: 11.1 g/day and the lowest salt-consuming region, Hokkaido/ Shikoku: 9.5 g/day), indicating that the regional differences in salt intake are becoming smaller [33]. This may be due to the influence of social development and the westernization of the Japanese diet [32].

The members of the WHO have committed to reducing the salt intake by 30% by 2025 [34]. Sustainable Development Goal Number 3 states that premature mortality from non-communicable diseases will be reduced to one-third by 2030 [35]. Each country has set out policies for the food service industry and processed food manufacturers to reduce the salt content of their products. In the United Kingdom, the Salt Reduction Program has been led by the Department of Health since 2003. This program has reduced salt intake from 9.5 g/day in 2000–2001 to 8.1 g/day in 2008 [36]. In 2018, the Spanish Agency for Consumer Affairs, Food and Nutrition released the Collaboration Plan for the Improvement of the Composition of Food and Beverages, which committed to reducing salt content in various food categories [37]. In Japan, the Health Japan 21 (second term) program aims at curtailing salt intake to 8 g/day [38]; however, there is no policy for the food industry regarding salt reduction. Although average salt intake in Japan has decreased due to medical and administrative population policy approaches, it has remained almost unchanged since 2015. The National Health and Nutrition Survey in 2019 investigated the intention to improve dietary habits, and more than 30% of the respondents answered they did not intend to improve their dietary habits even if they were consuming over 8 g/day salt [6]. In addition to a population approach, incorporating an environmental approach, such as working with the food industry to reduce the amount of sodium in their products, is also important in managing salt intake [39,40]. The Japanese Society of Hypertension has begun to work on an environmental approach toward reducing salt intake, such as including a certification for food items with a low salt content [41]. Moreover, 13 academic societies in Japan have formed a consortium, and a certification system for healthy and nutritional meal patterns (common name: Smart Meal) was launched in December 2019 in Japan. Smart Meal aims at restricting salt intake to 3.0–3.5 g/meal [42], which is expected to bring in environmental benefits as well.

Wallace et al. estimated the effect of using glutamates to substitute the amount of sodium among certain food groups in America, and reported that doing so could have a modest effect on the salt intake of the whole American population [43]. Some studies on animal models reported that adding MSG was associated with higher energy intake and obesity, while clinical and epidemiological studies have been inconsistent regarding a relationship between MSG consumption and energy intake and obesity [44]: Masic et al. reported adding MSG increased immediate appetite but reduced subsequent test meal intake [45], and He et al. reported MSG consumption was positively associated with BMI [46]. The effect of MSG on appetite is currently unclear, thus the effect of salt reduction using MSG would be expected to be more beneficial even considering the effect on appetite. Utilizing low-salt soups using umami, regardless of sex, age, salt exposure environment in early childhood, and current salty taste preference, will be useful to work toward an environmental approach to reducing salt intake.

This study had a few limitations. First, the participants of this study were registered students being trained as dietitians and health seminar participants and hence, likely to have been a highly health conscious group, which might have influenced the results. In addition, the proportion of female participants was high, and the number of people in each of the 12 regional blocks was unequal. In order to further generalize the data, it will be necessary to conduct a survey of populations with a wide range of characteristics. Second, in this study, the salt exposure environment in early childhood was defined as the average salt intake by regional block at birth in 1980, 1990, and 2000. However, the actual salt intake of each individual might differ. Furthermore, the average salt intakes by regional block at birth in 1980, 1990, and 2000 were quoted from the results of the National Nutrition Survey, but the sodium amount calculation method differs in each survey [9]. However, when examining food composition data, it is desirable that using better technologies, a more optimal method for obtaining an accurate intake is selected, as the old data cannot accurately calculate the current intake. This is a drawback when long-term investigations are conducted. Finally, we applied the data for the year 1980 to participants born before 1980 because no data on salt intake at birth were available before the year of 1980. The Japanese salt intake in 1980 was 12.9 g/day, while in 1975 it was 13.5 g/day [9]. Therefore, the salt intake by regional block before 1980 was also considered higher than in 1980.

## 5. Conclusions

There was no difference in the effect of umami ingredients on palatability between average salt intake by the regional block at birth and salty taste preference. These findings suggest that adding an appropriate amount of umami ingredients can facilitate salt reduction while maintaining palatability, regardless of early childhood salt exposure environment and current salty preference. If an environment is created in which umami is effectively utilized to reduce salt intake, it could be useful in the prevention and management of hypertension, and might contribute to a reduction in the incidence and mortality of cardiovascular disease as well.

## Figures and Tables

**Figure 1 nutrients-13-00577-f001:**
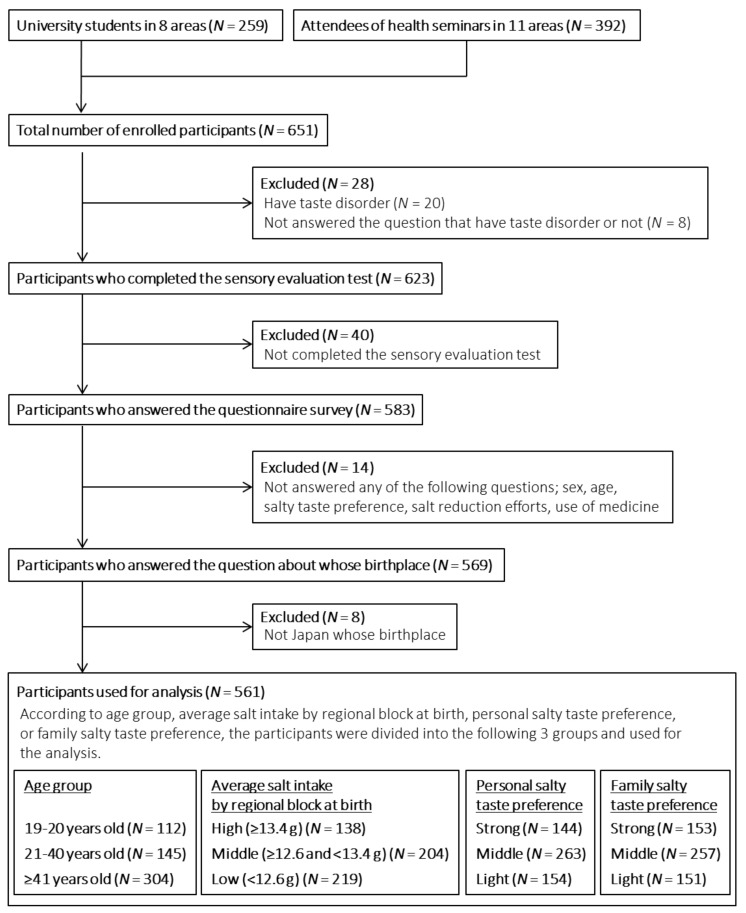
Flow chart on the analyzed participants.

**Table 1 nutrients-13-00577-t001:** The average amount and level of salt intake in 1980, 1990, and 2000 for 12 regional blocks in Japan along with the number of participants born in each regional block.

Regional Block ^†^	Name of Prefecture	1980			1990			2000		
		Salt Intake		*N* ^§^	Salt Intake		*N* ^¶^	Salt Intake		*N* ^††^
		Average (g/Day)	Level ^‡^		Average (g/Day)	Level ^‡^		Average (g/Day)	Level ^‡^	
Hokkaido	Hokkaido	14.4	H	2	12.7	M	1	12.3	L	2
Tohoku	Aomori, Iwate, Miyagi, Akita, Yamagata, Fukushima	15.8	H	33	13.5	H	7	13.8	H	19
Kanto-1	Saitama, Chiba, Tokyo, Kanagawa	12.7	M	22	12.5	L	32	12.6	M	16
Kanto-2	Ibaraki, Tochigi, Gunma, Yamanashi, Nagano	15.4	H	6	13.6	H	7	13.5	H	10
Hokuriku	Niigata, Toyama, Ishikawa, Fukui	14.2	H	6	12.8	M	5	12.8	M	2
Tokai	Gifu, Aichi, Mie, Shizuoka	11.8	L	27	12.1	L	17	12.4	L	22
Kinki-1	Kyoto, Osaka, Hyogo	10.9	L	10	11.8	L	6	11.6	L	3
Kinki-2	Nara, Wakayama, Shiga	11.5	L	9	13.4	H	4	11.2	L	1
Chugoku	Tottori, Shimane, Okayama, Hiroshima, Yamaguchi	12.3	L	27	12.5	L	14	11.9	L	14
Shikoku	Tokushima, Kagawa, Ehime, Kochi	12.0	L	1	12.3	L	2	12.0	L	1
Kita (Northern) Kyushu	Fukuoka, Saga, Nagasaki, Oita	13.0	M	121	11.6	L	9	11.6	L	18
Minami (Southern) Kyushu	Kumamoto, Miyazaki, Kagoshima, Okinawa	13.6	H	44	13.0	M	37	11.0	L	4
		the born-before-the 1980s group (*N*)		308	the born-in-the 1990s group (*N*)		141	the born-in-the 2000s group (*N*)		112

†: Regional blocks categorized by the National Nutrition Survey in Japan, and are listed in order from North to South. ‡: Participants are classified into three levels according to the average salt intake by regional block at birth (L: low <12.6 g, M: middle ≥12.6 and <13.4 g, H: high ≥13.4 g). §: Number of participants born in each regional block before 1989. ¶: Number of participants born in each regional block during 1990–1999. ††: Number of participants born in each regional block during 2000–2001.

**Table 2 nutrients-13-00577-t002:** Characteristics of the study participants according to sex and age.

	All *N* (%)		Sex				*p* Value ^‡^	Age						*p* Value ^§^
			Male *N* (%)		Female *N* (%)			19–20 *N* (%)		21–40 *N* (%)		≥41 *N* (%)		
All	561	(100)	64	(100)	497	(100)		112	(100)	145	(100)	304	(100)	
Age group (years)							<0.001							
19–20	112	(20.0)	5	(7.8)	107	(21.5)								
21–40	145	(25.8)	3	(4.7)	142	(28.6)								
≥41	304	(54.2)	56	(87.5)	248	(49.9)								
Salt intake by regional block at birth ^†^							<0.05							<0.001
High (≥13.4 g)	138	(24.6)	14	(21.9)	124	(24.9)		29	(25.9)	18	(12.4)	91	(29.9)	
Middle (≥12.6 and <13.4 g)	204	(36.4)	33	(51.6)	171	(34.4)		18	(16.1)	45	(31.0)	141	(46.4)	
Low (<12.6 g)	219	(39.0)	17	(26.6)	202	(40.6)		65	(58.0)	82	(56.6)	72	(23.7)	
Personal salty taste preference							<0.001							0.019
Strong	144	(25.7)	30	(46.9)	114	(22.9)		29	(25.9)	46	(31.7)	69	(22.7)	
Middle	263	(46.9)	21	(32.8)	242	(48.7)		60	(53.6)	68	(46.9)	135	(44.4)	
Light	154	(27.5)	13	(20.3)	141	(28.4)		23	(20.5)	31	(21.4)	100	(32.9)	
Family salty taste preference							0.818							<0.001
Strong	153	(27.3)	19	(29.7)	134	(27.0)		37	(33.0)	56	(38.6)	60	(19.7)	
Middle	257	(45.8)	27	(42.2)	230	(46.3)		57	(50.9)	58	(40.0)	142	(46.7)	
Light	151	(26.9)	18	(28.1)	133	(26.8)		18	(16.1)	31	(21.4)	102	(33.6)	
Salt reduction efforts							0.041							<0.001
Always	160	(28.5)	14	(21.9)	146	(29.4)		11	(9.8)	11	(7.6)	138	(45.4)	
Sometimes	247	(44.0)	24	(37.5)	223	(44.9)		46	(41.1)	82	(56.6)	119	(39.1)	
Rarely/Never	154	(27.5)	26	(40.6)	128	(25.8)		55	(49.1)	52	(35.9)	47	(15.5)	
Smoking habit							<0.001							<0.001
Current	14	(2.5)	7	(10.9)	7	(1.4)		2	(1.8)	2	(1.4)	10	(3.3)	
Former	28	(5.0)	20	(31.3)	8	(1.6)		0	(0)	0	(0)	28	(9.2)	
Never	519	(92.5)	37	(57.8)	482	(97.0)		110	(98.2)	143	(98.6)	266	(87.5)	
Medication							<0.001							<0.001
Yes	191	(34.0)	35	(54.7)	156	(31.4)		6	(5.4)	18	(12.4)	167	(54.9)	
No	370	(66.0)	29	(45.3)	341	(68.6)		106	(94.6)	127	(87.6)	137	(45.1)	

†: Participants are classified into three levels according to the average salt intake by 12 regional blocks in 1980, 1990, and 2000 from the National Nutrition Survey in Japan (low: <12.6 g, middle: ≥12.6 and <13.4 g, and high: ≥13.4 g). ‡: *p* for the comparisons between men and women, §: *p* for the comparisons between the three age groups (19–20 years, 21–40 years, and ≥41 years) with the use of chi-square tests.

**Table 3 nutrients-13-00577-t003:** Saltiness visual analogue scale (VAS) ratings by sex, age, salt intake by regional block at birth, and salty taste preference.

		0.3% NaCl		0.6% NaCl		0.9% NaCl		
	*N*	−MSG	+MSG	−MSG	+MSG	−MSG	+MSG	*p* Value ^‡^
All	561	23.51 ± 0.65 ^a^	39.28 ± 0.73 ^b^	55.81 ± 0.72 ^c^	58.24 ± 0.72 ^c^	73.76 ± 0.64 ^d^	72.42 ± 0.68 ^d^	<0.001
Sex								
Male	64	22.23 ± 1.84 ^a^	34.67 ± 2.12 ^b^	49.68 ± 2.51 ^c^	52.67 ± 2.04 ^c^	69.43 ± 1.85 ^d^	66.52 ± 1.97 ^d^	<0.001
Female	497	23.67 ± 0.69 ^a^	39.88 ± 0.77 ^b^	56.60 ± 0.73 ^c^	58.96 ± 0.76 ^c^	74.31 ± 0.68 ^d^	73.18 ± 0.71 ^d^	<0.001
*p* Value ^§^		0.567	0.581	0.259	0.518	0.182	0.548	
Age group (years)								
19–20	112	24.68 ± 1.48 ^a^	45.10 ± 1.41 ^b^	60.32 ± 1.37 ^c^	63.99 ± 1.17 ^c^	78.50 ± 1.06 ^d^	78.43 ± 1.08 ^d^	<0.001
21–40	145	23.68 ± 1.26 ^a^	45.54 ± 1.37 ^b^	61.89 ± 1.19 ^c^	66.77 ± 1.04 ^d^	79.23 ± 1.04 e	80.46 ± 0.86 e	<0.001
≥41	304	23.00 ± 0.88 ^a^	34.15 ± 0.96 ^b^	51.25 ± 1.01 ^c^	52.06 ± 1.02 ^c^	69.40 ± 0.94 ^d^	66.37 ± 0.98 ^d^	<0.001
*p* Value ^§^		0.304	<0.001	<0.001	<0.001	<0.001	<0.001	
Salt intake by regional block at birth ^†^								
High	138	22.04 ± 1.24 ^a^	38.67 ± 1.54 ^b^	53.66 ± 1.56 ^c^	54.14 ± 1.22 ^c^	71.43 ± 1.43 ^d^	67.57 ± 1.56 ^d^	<0.001
Middle	204	22.91 ± 1.11 ^a^	36.86 ± 1.12 ^b^	53.66 ± 1.12 ^c^	56.74 ± 1.34 ^c^	72.77 ± 1.02 ^d^	71.38 ± 1.06 ^d^	<0.001
Low	219	25.00 ± 1.03 ^a^	41.92 ± 1.18 ^b^	59.26 ± 1.11 ^c^	62.23 ± 1.01 ^c^	76.14 ± 0.98 ^d^	76.44 ± 0.97 ^d^	<0.001
*p* Value ^§^		0.238	0.706	0.242	0.104	0.534	0.008	
Personal salty taste preference								
Strong	144	19.45 ± 1.06 ^a^	37.72 ± 1.47 ^b^	54.37 ± 1.40 ^c^	57.40 ± 1.32 ^c^	73.55 ± 1.43 ^d^	71.94 ± 1.23 ^d^	<0.001
Middle	263	24.89 ± 0.94 ^a^	39.94 ± 1.09 ^b^	56.71 ± 1.07 ^c^	58.27 ± 1.09 ^c^	74.87 ± 0.92 ^d^	72.75 ± 1.02 ^d^	<0.001
Light	154	24.94 ± 1.38 ^a^	39.63 ± 1.29 ^b^	55.63 ± 1.32 ^c^	58.98 ± 1.35 ^c^	72.04 ± 1.10 ^d^	72.31 ± 1.31 ^d^	<0.001
*p* Value ^§^		0.002	0.127	0.408	0.153	0.420	0.465	
Family salty taste preference								
Strong	153	21.22 ± 1.15 ^a^	40.85 ± 1.35 ^b^	56.58 ± 1.17 ^c^	60.38 ± 1.30 ^c^	74.34 ± 1.32 ^d^	74.61 ± 1.17 ^d^	<0.001
Middle	257	25.24 ± 0.98 ^a^	38.90 ± 1.08 ^b^	55.21 ± 1.12 ^c^	57.03 ± 1.08 ^c^	73.67 ± 0.93 ^d^	71.56 ± 1.04 ^d^	<0.001
Light	151	22.88 ± 1.26 ^a^	38.33 ± 1.43 ^b^	56.06 ± 1.43 ^c^	58.15 ± 1.39 ^c^	73.30 ± 1.20 ^d^	71.67 ± 1.33 ^d^	<0.001
*p* Value ^§^		0.028	0.895	0.521	0.343	0.460	0.640	

Values are means ± standard errors (SEs) One underline is for the highest ratings in the 0.3%, 0.6%, and 0.9% NaCl solutions alone. Double underline is for the highest ratings in the 0.3%, 0.6%, and 0.9% NaCl solutions with MSG. The significant differences between each rating are indicated by alphabetic superscripts. A rating is significantly different from others that have different superscript letters according to Tukey’s test (*p* < 0.05). †: Participants are classified into three levels according to the average salt intake by 12 regional blocks in 1980, 1990, and 2000 from the National Nutrition Survey in Japan (low: <12.6 g, middle: ≥12.6 and <13.4 g, and high: ≥13.4 g). ‡: *p* for repeated measures ANOVA. §: *p* for ANCOVA adjusted for sex, age group (19–20 years, 21–40 years, and ≥41 years), smoking habit (current, former, and never), salt reduction efforts (always, sometimes, and rarely/never), and use of medicine (yes and no).

**Table 4 nutrients-13-00577-t004:** Palatability VAS ratings by sex, age, salt intake by regional block at birth, and salty taste preference.

		0.3% NaCl		0.6% NaCl		0.9% NaCl		
	*N*	−MSG	+MSG	−MSG	+MSG	−MSG	+MSG	*p* Value ^‡^
All	561	30.66 ± 0.78 ^a^	60.95 ± 0.90 ^b^	44.87 ± 0.71 ^c^	57.62 ± 0.75 ^d^	40.76 ± 0.75 ^e^	51.81 ± 0.83 ^f^	<0.001
Sex								
Male	64	29.18 ± 2.34 ^a^	56.41 ± 2.58 ^b^	39.81 ± 2.04 ^c^	55.16 ± 1.75 ^b^	41.52 ± 2.11 ^cd^	50.20 ± 2.48 ^bd^	<0.001
Female	497	30.85 ± 0.83 ^a^	61.54 ± 0.96 ^b^	45.52 ± 0.75 ^c^	57.94 ± 0.82 ^d^	40.66 ± 0.80 ^e^	52.01 ± 0.89 ^f^	<0.001
*p* Value ^§^		0.695	0.273	0.078	0.409	0.544	0.319	
Age group (years)								
19–20	112	32.48 ± 1.49 ^a^	59.57 ± 1.86 ^b^	46.36 ± 1.39 ^cd^	56.58 ± 1.65 ^be^	41.44 ± 1.49 ^c^	51.53 ± 1.97 ^de^	<0.001
21–40	145	31.60 ± 1.73 ^a^	63.66 ± 1.88 ^b^	48.45 ± 1.44 ^cd^	60.16 ± 1.54 ^b^	43.35 ± 1.48 ^c^	51.47 ± 1.76 ^d^	<0.001
≥41	304	29.54 ± 1.05 ^a^	60.17 ± 1.22 ^b^	42.61 ± 0.97 ^c^	56.80 ± 1.00 ^b^	39.27 ± 1.06 ^c^	52.07 ± 1.07 ^d^	<0.001
*p* Value ^§^		0.154	0.147	0.040	0.230	0.175	0.399	
Salt intake by regional block at birth ^†^								
High	138	32.70 ± 1.67 ^a^	59.33 ± 1.69 ^b^	46.34 ± 1.43 ^cd^	57.48 ± 1.62 ^be^	41.29 ± 1.56 ^c^	52.69 ± 1.68 ^de^	<0.001
Middle	204	28.19 ± 1.20 ^a^	61.60 ± 1.56 ^b^	43.42 ± 1.24 ^c^	57.11 ± 1.24 ^b^	39.60 ± 1.29 ^d^	51.45 ± 1.32 ^c^	<0.001
Low	219	31.67 ± 1.27 ^a^	61.37 ± 1.45 ^b^	45.29 ± 1.07 ^c^	58.19 ± 1.15 ^b^	41.50 ± 1.15 ^c^	51.58 ± 1.39 ^d^	<0.001
*p* Value ^§^		0.099	0.642	0.246	0.877	0.743	0.845	
Personal salty taste preference								
Strong	144	28.23 ± 1.40 ^a^	61.40 ± 1.79 ^b^	45.04 ± 1.36 ^c^	58.73 ± 1.31 ^bd^	43.34 ± 1.49 ^c^	55.26 ± 1.53 ^d^	<0.001
Middle	263	30.75 ± 1.13 ^a^	61.92 ± 1.31 ^b^	44.64 ± 1.05 ^c^	58.99 ± 1.09 ^b^	40.34 ± 1.13 ^c^	52.17 ± 1.22 ^d^	<0.001
Light	154	32.77 ± 1.62 ^a^	58.88 ± 1.72 ^b^	45.09 ± 1.37 ^cd^	54.26 ± 1.57 ^b^	39.06 ± 1.34 ^c^	47.97 ± 1.65 ^d^	<0.001
*p* value ^§^		0.085	0.624	0.702	0.082	0.317	0.007	
Family salty taste preference								
Strong	153	30.29 ± 1.54 ^a^	61.77 ± 1.70 ^b^	47.07 ± 1.19 ^cd^	58.26 ± 1.36 ^b^	43.35 ± 1.43 ^c^	51.77 ± 1.57 ^d^	<0.001
Middle	257	30.87 ± 1.11 ^a^	60.89 ± 1.31 ^b^	44.32 ± 1.08 ^c^	57.88 ± 1.11 ^b^	41.13 ± 1.11 ^c^	52.45 ± 1.23 ^d^	<0.001
Light	151	30.67 ± 1.58 ^a^	60.22 ± 1.81 ^b^	43.57 ± 1.45 ^c^	56.54 ± 1.53 ^bd^	37.51 ± 1.44 ^c^	50.75 ± 1.66 ^d^	<0.001
*p* Value ^§^		0.843	0.989	0.497	0.935	0.096	0.830	

Values are means ± standard errors (SEs). One underline is for the highest ratings in the 0.3%, 0.6%, and 0.9% NaCl solutions alone. Double underline is for the highest ratings in the 0.3%, 0.6%, and 0.9% NaCl solutions with MSG. The significant differences between each rating are indicated by alphabetic superscripts. A rating is significantly different from others that have different superscript letters according to Tukey’s test (*p* < 0.05). †: Participants are classified into three levels according to the average salt intake by 12 regional blocks in 1980, 1990, and 2000 from the National Nutrition Survey in Japan (low: <12.6 g, middle: ≥12.6 and <13.4 g, and high: ≥13.4 g). ‡: *p* for repeated measures ANOVA. §: *p* for ANCOVA adjusted for sex, age group (19–20 years, 21–40 years, and ≥41 years), smoking habit (current, former, and never), salt reduction efforts (always, sometimes, and rarely/never), and use of medicine (yes and no).

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
