# Peer review of "Effect of Monosodium Glutamate on Saltiness and Palatability Ratings of Low-Salt Solutions in Japanese Adults According to Their Early Salt Exposure or Salty Taste Preference"

_nutrients, 2021, doi:10.3390/nu13020577_

Round 1
Reviewer 1 Report
The article investigates the effect of MSG on the saltiness and owerall palatability of low-salt solutions in Japanese participants, being the birthplace of participants the only new considered data compared to a previous paper of the same group.
There is a huge difference in the gender of considered subjects (64 males, 497 females), as quoted by the Authors in the limitation of the work.
English could be improved.
I think overall the paper do not add much to what already known in the scientific society.

Reviewer 2 Report
This work is of relevance to the field of enhancement of taste in relation to saltiness with importance. In the current manuscript the authors investigated the possibility that the salt exposure environment in early childhood can be influence current salty taste preference in adult and the relation between umami and saltiness.
The novelty of the manuscript was low as authors pointed in introduction that there already are many studies on the effect of umami taste compounds on saltiness, consequently reduce salt consumption. It is hard to find out what is new finding in this research from others? Authors should be focus to explain their main purpose, results, and logically discuss what new findings from this research are.
Authors mentioned in introduction (L71-L73) “In this study, we focused on the differences between salt exposure environment in early childhood and current salty taste preference…..” However, it is not clearly summarized about this main purpose of the research and results in the Abstract. The abstract should be rephrased and the important data should be added.
The introduction should be revised and more related studies should be added, particularly on relations between salt and umami taste.
Some experimental details are mentioned in introduction L62-L66 and also mentioned in the Materials & Methods, 2.1 Study Design. This should be deleted from introduction.
On experimental design, there is too much difference in numbers between Male and female. Is there no problem to translate their data?
Add the reason why authors employed three salt concentrations, 0.3%, 0.6%, and 0.9% in this study and the reason why caused different results depend on the concentrations.
Authors discussed their result controversially as below. To support these results, add discussion with references that there is no relation on sensitivity and palatability on saltiness.“This means that those who were exposed to low levels of salt in early childhood were more sensitive to salty taste than those exposed to high salt levels.”(L261-262) Then,“Our results, however, showed that early childhood salt exposure did not affect palatability.”(L280-283)
Taken overall, it is very hard to read this manuscript. Rephrase or Tables or use Figure may help to understand the results.
Reviewer 3 Report
The article of Morita et al. explores potential avenues for salt reduction in Japanese population, a novel and a topic of public health interest. More specifically, the authors explore the effect of monosodium L-glutamate (MSG) on the saltiness and palatability of low-salt solutions. The results show that palatability was higher with MSG than without MSG for each NaCl solution, with a peak value for the 0.3% NaCl solution with MSG. There was no effect of average of salt intake by regional block at birth and salty preference. The authors conclude that adding an appropriate amount of umami ingredients can facilitate salt reduction in diet while maintaining palatability regardless of salt-exposed environment in early childhood or salty preference.
The article is overall well written with clear methods, results and a well written discussion. The authors also acknowledge the limitations of their study such as generalisability and sodium intake measurement.
I have a few minor points that require clarification:
- Th authors state the salty taste preference was measured by asking questions about the degree of personal salty taste preference (very light, light, middle, strong, very strong). Has this been used before in other research?
- Statistical analysis section: please state the test of normality used.
- Line 220-222: the authors state “and each of these solutions had a lower NaCl concentration than the solutions without MSG” - this should be explained further as it is not entirely clear until reading the discussion
Round 2
Reviewer 1 Report
After the revision the papers reads a bit more smoothly.
But the main point (difference in number of male/female judges) could not be changed.
I do not think the paper add much to what already published on the subject.
Reviewer 2 Report
It is improved in this revision.
Author Response
We appreciate your kind words and wish to reiterate that your insightful comments on our paper helped us significantly improve it.